# Tree Species Classification Using Optimized Features Derived from Light Detection and Ranging Point Clouds Based on Fractal Geometry and Quantitative Structure Model

Zhenyang Hui [1,2], Zhaochen Cai [1,2], Peng Xu [3], Yuanping Xia [1,2,*] and Penggen Cheng [1,2]

1   Key Laboratory of Mine Environmental Monitoring and Improving around Poyang Lake of Ministry of Natural Resources, East China University of Technology, Nanchang 330013, China; huizhenyang2008@ecut.edu.cn (Z.H.); 2021110139@ecut.edu.cn (Z.C.); pgcheng@ecut.edu.cn (P.C.)
2   School of Surveying and Geoinformation Engineering, East China University of Technology, Nanchang 330013, China
3   Powerchina Guiyang Engineering Corporation Limited, Guiyang 550081, China; xup_gyy@powerchina.cn
*   Correspondence: ypxia@ecut.edu.cn

**Abstract:** Tree species classification is a ubiquitous task in the forest inventory field. Only directly measured feature vectors have been applied to most existing methods that use LiDAR technology for tree species classification. As a result, it is difficult to obtain a satisfactory tree species classification performance. To solve this challenge, the authors of this paper developed two new kinds of feature vectors, including fractal geometry-based feature vectors and quantitative structural model (QSM)-based feature vectors. In terms of fractal geometry, both two fractal parameters were extracted as feature vectors for reflecting how tree architecture is distributed in three-dimensional space. In terms of QSM, the ratio of length change and the ratio of radius change of different branches were extracted as feature vectors. To reduce the feature vector dimensionality and explore valuable feature vectors, feature vector dimension reduction was conducted using the classification and regression tree (CART). Five hundred and sixty-eight individual trees with five tree species were selected for evaluating the performance of the developed feature vectors. The experimental results indicate that the tree species of Fagus sylvatica achieved the highest overall accuracy, which is 98.06%, while Quercus petraea obtained the lowest overall accuracy, which is 96.65%. Four other classical supervised learning methods were adopted for comparison. The comparison result indicates that the proposed method outperformed the other four supervised learning methods no matter which accuracy indicator was adopted. In comparison with the relevant method, the eight feature vectors developed in this paper also performed much better. This indicates that the fractal geometry-based feature vectors and QSM-based feature vectors developed in this paper can effectively improve the performance of tree species classification.

**Keywords:** tree species; LiDAR point cloud; feature vectors; fractal geometry; quantitative structure model

## 1. Introduction

Forests are one of the most important ecosystems on earth and are of great significance to economic construction, climate change and human survival [1–3]. The composition of forest tree species has a direct impact on ecological attributes, such as the greenhouse gas absorption capacity and the resource utilization rate. The richness of forest tree species can also reflect the level of forest productivity and forest biodiversity [4,5]. Therefore, tree species classification has become a ubiquitous task in forest resource inventory, and also, a popular issue in current research [4,6,7].

LiDAR is an active remote sensing technology. The laser pulses emitted by an LiDAR system can not only obtain the three-dimensional coordinate information about a target,

but also penetrate vegetation and accurately depict spatial structure information about the vegetation canopy [8–10]. In recent years, with the rapid development of LiDAR technology, its sampling rate and sampling accuracy have obviously been improved, making LiDAR technology widely used in the forest inventory field [11].

Numerous tree species identification methods using LiDAR technology have been proposed in recent years. Lin and Herold [12] proposed explicit tree structure (ETS) feature parameters derived from using terrestrial laser scanning (TLS) to classify boreal tree species. In their method, there are five types of ETS feature parameter, including the structural characteristics of an entire tree, stem, branches, crown and leaves. And then, leave-one-out-cross-validation was adopted in SVM to evaluate the performance of extracted ETS feature parameters. The experimental results showed that the average and maximum classification accuracies for 40 samples of four typical boreal tree species were up to 77.5% and 90%, respectively. In the method proposed by Åkerblom et al. [6], they first used the TreeQSM method proposed by Raumonen et al. [13] to construct a quantitative structure model (QSM) of tree point clouds. Fifteen types of features were extracted from the QSM, and then optimized based on the performance of different feature sets in their method. The experimental results showed that using ten optimized features can achieve a maximum classification accuracy of 96.9%, which proved that geometric features extracted from QSM can effectively achieve tree species classification. Terryn et al. [2] proposed two new feature parameters, including the branch angle ratio and relative volume, on the basis of the study proposed by Åkerblom et al. [6]. Seven hundred and fifty-eight trees of five species (Acer pseudoplatanus, Fraxinus excelsior, Crataegus monogyna, Corylus avellana and Quercus robur) in a 1.4-hectare mixed deciduous forest site were classified. They achieved about 80% overall accuracy for the experimental sample after using principal component analysis on the feature parameters. Xi et al. [4] extracted 32 classification features in total after the QSM was constructed using the method proposed by Xi et al. [14]. In their experiments, the classification performances of seven deep learning and six machine learning classifiers were compared. Among them, the PointNet++ method achieved the best overall classification accuracy, and compared to other deep learning methods, the training time of this method was also the shortest. However, their experimental results also demonstrated that when there are a few tree samples, this method is prone to overfitting, which will reduce the classification accuracy. Liu et al. [15] constructed a deep neural network learning model named LayerNet for the species classification of LiDAR tree point clouds in simple forest areas. They validated the classification performance of this method using both airborne and ground-based LiDAR datasets. The experimental results showed that their proposed method can achieve overall classification accuracies of 88.8% and 92.5% for airborne and ground-based datasets, respectively. Their method can also achieve more satisfactory accuracy and Kappa coefficients in comparison to those of traditional machine learning and deep learning methods. However, the information redundancy caused by large feature dimensions will affect the performance of this method.

In addition to the tree species classification methods using LiDAR point clouds, some researchers also tried to fuse LiDAR data and other data sources to obtain tree species classification results. Dalpont et al. [16] investigated the effect of combining hyperspectral data with LiDAR data for classification in complex forest environments with more than 19 tree species. Their study showed that combining hyperspectral data with LiDAR data could improve the tree species identification performance in cases where the spectral information were relatively similar, but the tree heights obviously differed between species. Kim et al. [17] further investigated the tree species identification performance using multiple LiDAR intensity data. They used intensity information derived from leaf-on and leaf-off point clouds of trees in the same forest site to identify coniferous and broadleaved species. The experimental results showed that when leaf-off point clouds were used, the classification accuracy was higher than that when leaf-on point clouds were used, and the highest accuracy of 90.06% was obtained when the combination of both leaf-on and leaf-off point clouds were used. Puttonen et al. [18] combined hyperspectral data and tree

shape features from LiDAR to form a fused dataset. And then, the fused dataset was used to classify 24 tree samples of three tree species (Birch, Norway spruce and Scots pine) in the SVM. The classification performances using the fused datasets and two single source datasets (hyperspectral data and tree shape features) were compared. The experimental results showed that the best classification performance was realized when the fused dataset was used, which was able to achieve an accuracy above 85%. Othmani et al. [19] classified tree species using the three-dimensional texture feature of bark. In their method, a two-dimensional image representing the three-dimensional geometric texture feature of the bark at around breast height was generated first. Next, the multiresolution analysis technique was applied to the image to extract the texture feature. The extracted features can achieve a classification accuracy up to 86.93% for experimental tree samples, but this method had struggles to recognize Pine and Hornbeam species. Zhang et al. [20] proposed a method that combined canopy information derived from LiDAR and spectral information in hyperspectral imagery to classify urban tree species. In their method, the canopy height model derived from LiDAR data was segmented via object-based image analysis to obtain individual tree crowns. Then, significant bands of individual tree spectrums were selected via minimum noise fraction transformation, and the classification performance of the selected bands was measured using random forest and multi class classifiers.

Although fusing LiDAR point clouds with other remote sensing data, such as hyperspecrtal image, can improve the performance of tree species classification, the fusion process is prone to error. Realizing the high-precision fusion of multi-source data is still a challenge. Thus, the classification of different tree species using LiDAR point clouds alone is still conducted by researchers. However, recognizing tree species from LiDAR point clouds still suffers from the following two problems. One is how to explore more effective features for tree species classification. The other one is how to obtain a combination of the feature vectors to achieve a higher classification accuracy, while reducing the feature vector dimension. To solve these challenges, this paper proposes a tree species classification method based on the combination of developed fractal geometry-based features and QSM-based features. In this method, three kinds of feature vectors were explored, including directly measured feature vectors, fractal geometry-based feature vectors and QSM-based feature vectors. In this paper, fifteen feature vectors in total were calculated. To reduce feature vector dimension, the classification and regression tree (CART) was adopted to analyze the importance of each feature vector. Among the fifteen feature vectors, ten important feature vectors were extracted for further analysis. Thereafter, to further reduce feature vector dimension, eight feature vectors with a high accuracy level and high occurrence frequency were extracted. The support vector machine (SVM) classification method was applied for tree species classification using the final eight extracted feature vectors. The experimental results show that the proposed method can obtain satisfying tree species classification results.

## 2. Materials and Methods

### 2.1. Datasets

This paper adopted the individual tree point clouds provided by Weiser et al. [21] for evaluating the performance of tree species classification. In this dataset, twelve forest plots of approximately 1 ha were scanned using laser scanners of different platforms, including airborne LiDAR, unscrewed aerial vehicle (UAV)-based LiDAR and terrestrial LiDAR. The twelve forest plots are located in mixed central European forests close to Bretten and Karlsruhe, in the federal state of Baden-Württemberg, Germany. This dataset provided segmented individual tree point clouds from different platforms, corresponding tree species information and tree metrics. Since the feature vectors developed in this paper need to construct the QSM, only the individual tree point clouds of high quality obtained via UAV-based LiDAR and terrestrial LiDAR were selected for testing. Moreover, to balance the number of samples of different tree species, some tree species with smaller number of trees were ignored. Eventually, 568 individual trees of five tree species, such as Fagus

sylvatica, Picea abies, Pinus sylvestris, Pseudotsuga menziesii and Quercus petraea, were selected for evaluating the performance of the developed feature vectors. The detailed information for the selected individual trees is tabulated in Table 1. Figure 1 shows several individual trees of the five tree species. It can be found that all these individual trees own high-quality point clouds. Thus, the QSM of each individual tree can be accurately built. Meanwhile, several tree metrics, such as *DBH*, can be precisely calculated.

**Table 1.** Information about selected individual trees. Std represents the standard deviation. abbr means abbreviation.

| Species | Number of Trees | Average of Height (m) | Std of Tree Height (m) | Average of *DBH* (m) | Std of *DBH* (m) |
|---|---|---|---|---|---|
| *Fagus sylvatica* (abbr: FagSyl) | 129 | 28.02 | 4.08 | 0.35 | 0.15 |
| *Picea abies* (abbr: PicAbi) | 123 | 20.84 | 5.59 | 0.26 | 0.12 |
| *Pinus sylvestris* (abbr: PinSyl) | 81 | 29.20 | 3.45 | 0.29 | 0.13 |
| *Pseudotsuga menziesii* (abbr: PseMen) | 124 | 36.30 | 4.76 | 0.28 | 0.12 |
| *Quercus petraea* (abbr: QuePet) | 111 | 21.18 | 7.46 | 0.19 | 0.08 |

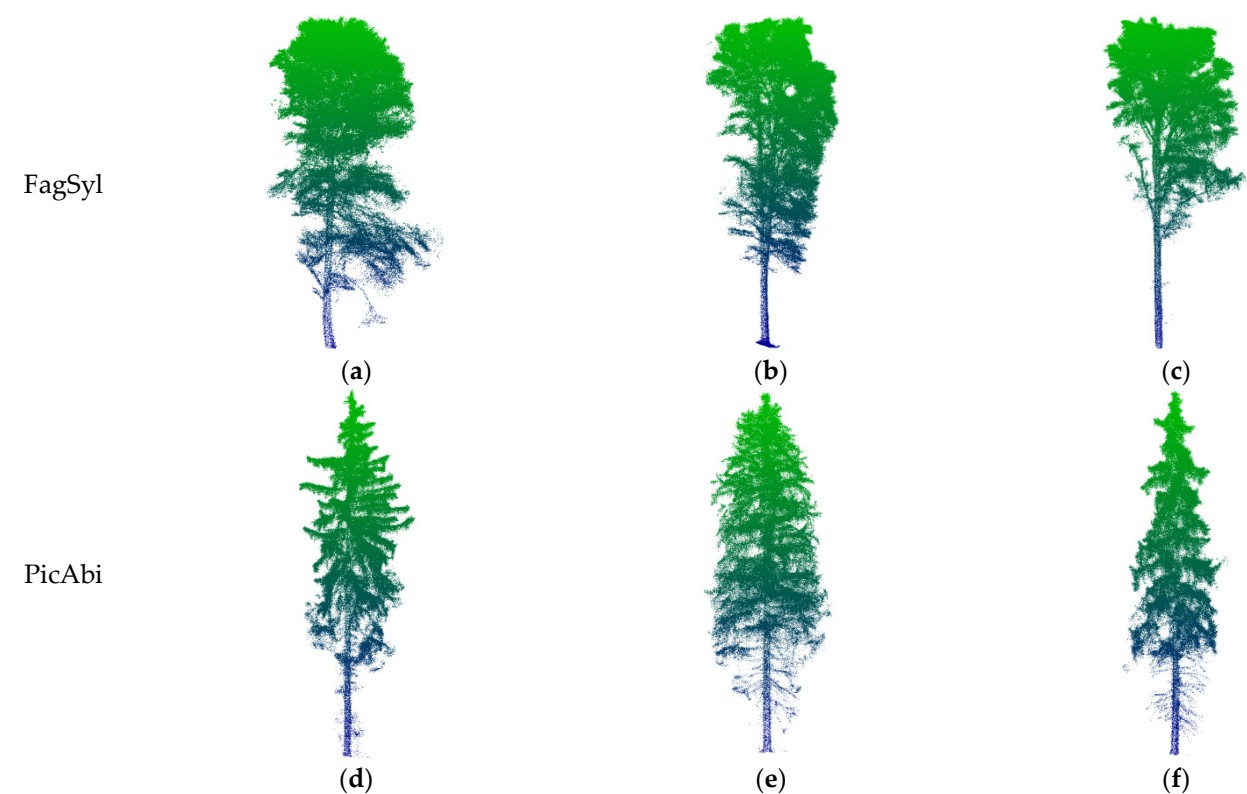

FagSyl

(**a**)　　　　　(**b**)　　　　　(**c**)

PicAbi

(**d**)　　　　　(**e**)　　　　　(**f**)

**Figure 1.** *Cont*.

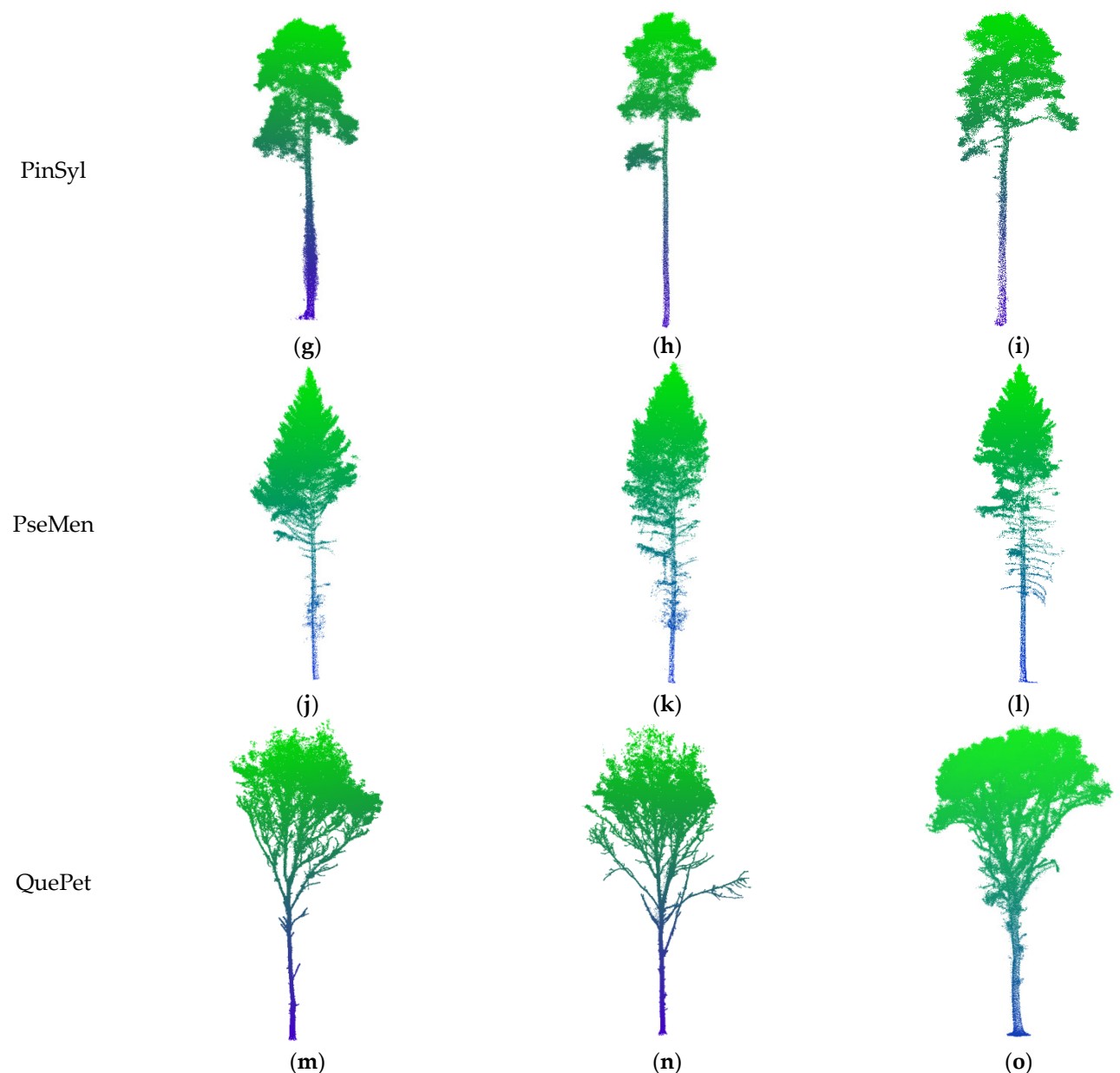

PinSyl

(g)  (h)  (i)

PseMen

(j)  (k)  (l)

QuePet

(m)  (n)  (o)

**Figure 1.** Samples of individual trees of different tree species. (**a**–**c**) are samples of tree species of FagSyl; (**d**–**f**) are samples of tree species of PicAbi; (**g**–**i**) are samples of tree species of PinSyl; (**j**–**l**) are samples of tree species of PseMen and (**m**–**o**) are samples of tree species of QuePet.

Figure 2a,b shows the distribution of tree heights and DBHs of all the individual trees tabulated in Table 1. It can be found that the height of trees varies greatly from species to species. Combined with Table 1, the average of height of different tree species changes from 20.84 m to 36.30 m. In terms of *DBH*, the average values of *DBH* of these five species do not change greatly. From Table 1, it can also be found that the smallest average value of DBH is 0.19 m, while the largest one is 0.35 m.

### 2.2. Method

The flowchart of the proposed method is shown in Figure 3. In the proposed method, multi-dimensional feature vectors were first extracted from each individual tree points. And then, the dimensionality reduction of feature vectors was applied for reducing the computation burden. Subsequently, the final eight feature vectors were obtained based on the importance and occurrence frequency of each feature vector. Lastly, support vector

machines (SVM) were adopted to obtain the final tree species classification results. Three main steps are involved in this paper, namely: (i) multi-dimensional feature vectors extraction, (ii) the dimensionality reduction of feature vectors, and (iii) the selection of feature vectors combination.

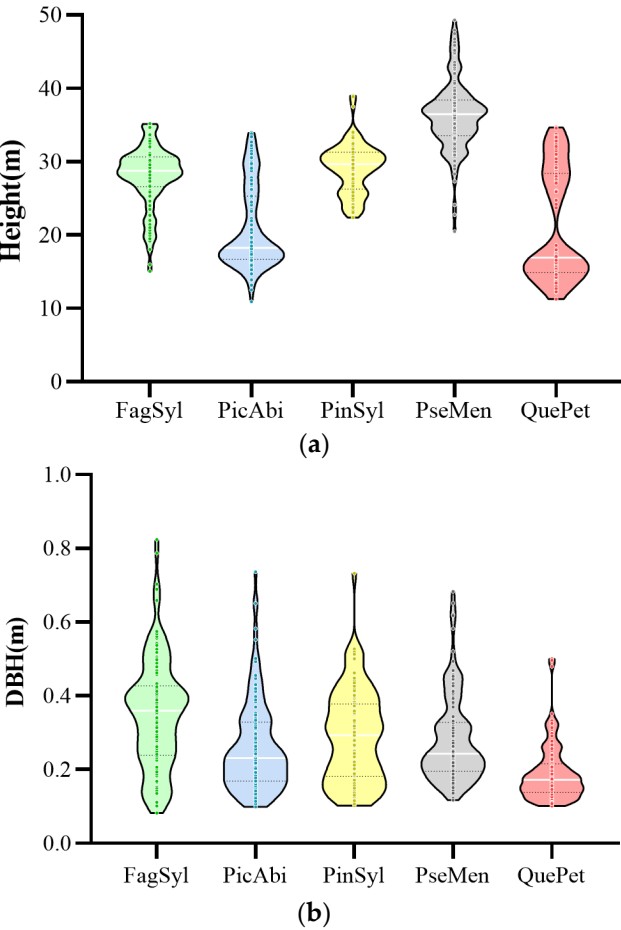

**(a)**

**(b)**

**Figure 2.** Distribution of tree heights and *DBH*s of the five tree species: (**a**) tree height distribution of different tree species; (**b**) *DBH* distribution of different tree species. In the figure, the white dotted line represents median, while the two black dotted lines represent 25 percentile and 75 percentile, respectively.

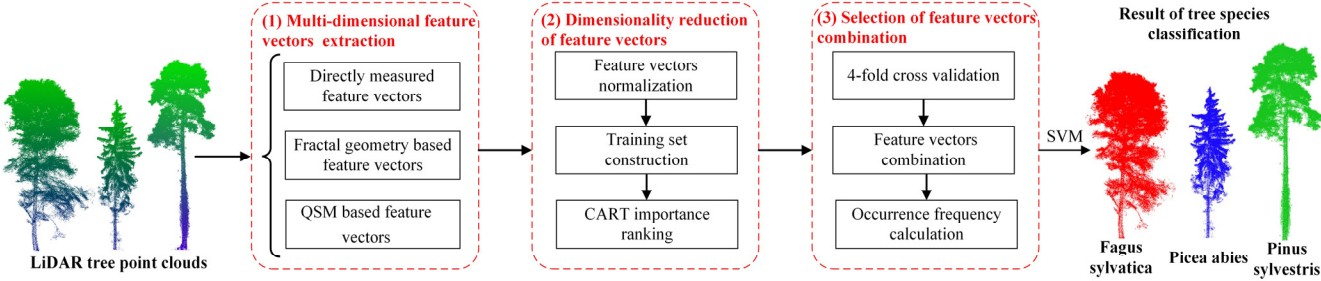

**Figure 3.** Flowchart of the proposed method.

### 2.2.1. Multi-Dimensional Feature Vectors Extraction

In this method, three kinds of feature vectors were developed, including directly measured feature vectors, fractal geometry-based feature vectors and QSM-based feature vectors. These three kinds of feature vectors are shown in Figure 4.

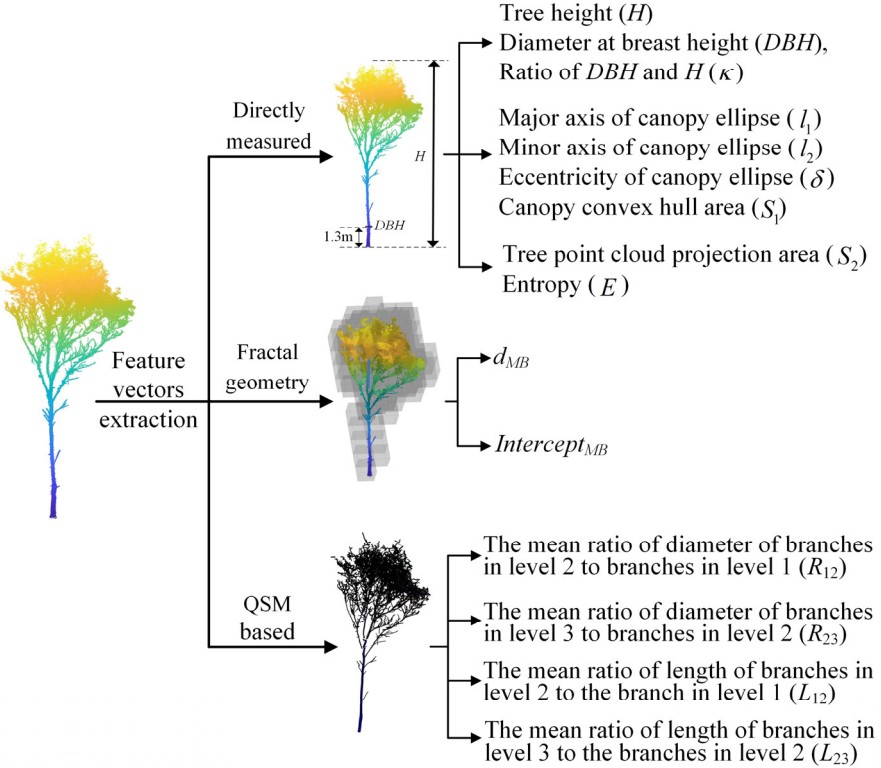

**Figure 4.** Multi-dimensional feature vectors extraction.

**(i) Directly measured feature vectors**

The first kind of feature vectors can be directly measured from the individual tree point cloud, including the tree height ($H$), diameter at breast height ($DBH$), ratio of $DBH$ and $H$ ($\kappa$), major axis of canopy ellipse ($l_1$), minor axis of canopy ellipse ($l_2$), eccentricity of canopy ellipse ($\delta$), canopy convex hull area ($S_1$), tree point cloud projection area ($S_2$) and entropy ($E$). $H$ is calculated as the elevation difference between the highest and lowest points within one individual tree. $DBH$ is the tree diameter at 1.3 m from the tree's root. In this paper, $DBH$ was calculated using the method proposed by Di Wang et al. [22]. $l_1$ and $l_2$ are calculated as the major and minor axes of the fitted ellipse for canopy points. $\delta$ is the corresponding eccentricity of the fitted ellipse. $S_1$ is calculated as the convex hull area of canopy point. $S_2$ is the horizontal projection area of all tree points, which can be calculated by gridding all the points with x and y coordinates. $E$ can be calculated via voxelizing the tree points. The ratio of the number of points within each voxel to the number of all tree points can be calculated as $p_i$. $E$ is defined as Equation (1):

$$E = -\sum_{i=1}^{n} p_i \times \log(p_i) \tag{1}$$

where $n$ is the number of voxels.

**(ii) Fractal geometry-based feature vectors**

Fractal geometry considers that many objects have the hierarchy of self-similarity, which can be observed at different scales. When the geometry is appropriately scaled up or down, the whole structural feature does not change. Guzman Q et al. [23] have proven that there is a noticeable correlation between the fractal geometry parameters and tree metrics, such as tree height, DBH and crown area. Since these tree metrics can be used for tree species identification, the authors of this paper tried to apply geometry parameters for classifying different tree species.

The fractal geometry parameters can be calculated based on the box counting method. This method considers that the individual tree points can be covered by a series of boxes, as

shown in Figure 5a. When the voxel size changes from large to small, the number of voxels for covering the tree points will be distinctly increased. A log–log linear regression model can be built using the voxel size and the number of voxels, as defined in Equation (2):

$$\log N = d_{MB} \times \log \frac{1}{V} + Intercept_{MB} \tag{2}$$

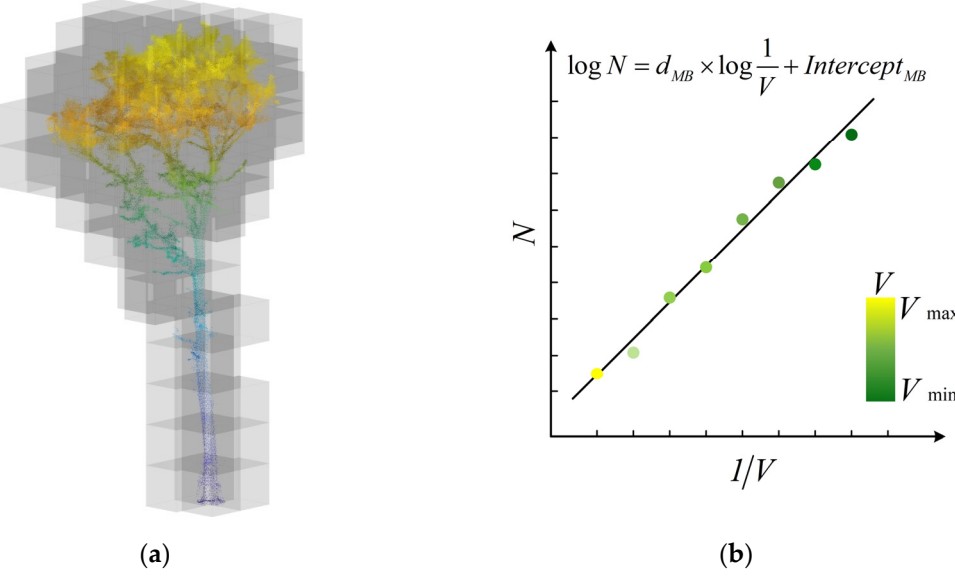

**Figure 5.** Fractal geometry parameters calculation using the box counting method: (**a**) tree points covered with a series of boxes; (**b**) a log–log linear regression using voxel size (*V*) and the number of voxels (*N*).

As shown in Figure 5b, $d_{MB}$ is the slope of the linear regression equation, which is defined as the fractal dimension. $Intercept_{MB}$ is the intercept of the linear model, which is defined as the fractal intercept. Both these two fractal parameters were used for tree species classification in this paper.

**(iii) QSM-based feature vectors**

The QSM can reflect the structure features and spatial topology of an individual tree. Thus, the authors of this paper tried to extract feature vectors based on the QSM for tree species classification. In this paper, the QSM was constructed using the TreeQSM method proposed by Raumonen et al. [13]. In TreeQSM, the model of an individual tree is constructed using a series of fitted cylinders. Moreover, the topology of each tree branch can also be built via TreeQSM. That is, TreeQSM can separate branches into different levels. In general, different tree species have different ratios of length and diameter of branches at different levels, such as level 1, level 2, level 3, etc., as shown in Figure 6. Thus, this paper developed four feature vectors for tree species classification, as defined in Equations (3)–(6):

$$R_{12} = \sum_{i=1}^{K} \left( \frac{R_2^i}{R_1^i} \right) / K \tag{3}$$

$$R_{23} = \sum_{j=1}^{M} \left( \frac{R_3^j}{R_2^j} \right) / M \tag{4}$$

$$L_{12} = \sum_{i=1}^{K} \left( \frac{L_2^i}{L_1^i} \right) / K \tag{5}$$

$$L_{23} = \sum_{j=1}^{M} \left( \frac{L_3^j}{L_2^j} \right) / M \tag{6}$$

where $R_{12}$ and $R_{23}$; $L_{12}$ and $L_{23}$ represent the ratio of radius and the ratio of length of branches at different levels, respectively.

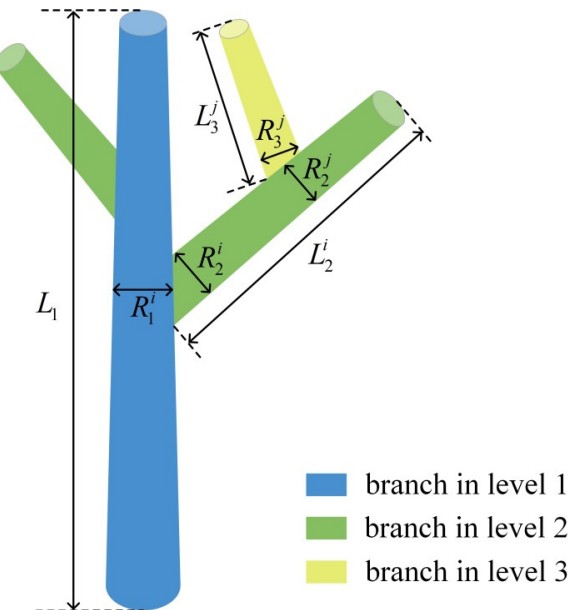

**Figure 6.** The sketch map of branches of different levels in QSM.

2.2.2. Dimensionality Reduction of Feature Vectors

As mentioned above, these three kinds of feature vectors contain fifteen feature vectors in total. Although fifteen feature vectors were developed in this paper, not all the feature vectors have noticeable contributions to tree species identification. To reduce feature vector dimensionality, this paper further analyzes the relative importance of each feature vector using the classification and regression tree (CART). The CART can be used to judge the importance of each feature vector in terms of classification and further eliminate the relatively low level of importance of classification features to achieve dimensionality reduction (Zhouxin Xi et al. [4]). In the CART method, the Gini index was adopted for selecting partition attribute when the decision tree was built, which is defined as Equation (7).

$$Gini(D) = 1 - \sum_{k=1}^{N} p_k^2 \tag{7}$$

where $p_k$ is the proportion of class $k$. *Gini* $(D)$ reflects the probability that two samples are randomly selected from the dataset, $D$, and their categories are inconsistent. Obviously, the smaller $Gini(D)$ is, the more purity the dataset $D$ is. When building the CART decision tree, the leaf node corresponds to the decision result, while the branch nodes correspond to an attribute being split. In this paper, the branch nodes represent attributes of these feature vectors for splitting. Ideally, when a decision tree is built, the samples contained by the branch nodes should belong to the same category as far as possible. This means that the purity of the nodes increases. On the contrary, at each node, the risk of splitting is estimated to be the node impurity. The authors of this paper calculated the importance of each feature vector by summing changes in the risk due to splits on each node.

The relative importance of each vector calculated using the CART is shown in Figure 7. It can be found that although there are nine feature vectors involved in the directly measured feature vectors, three of them show a relatively low level of importance for tree species

identification. These three feature vectors are *DBH*, the ratio ($\kappa$) of *DBH* and *H* and the eccentricity ($\delta$) of canopy ellipse. In terms of fractal geometry-based feature vectors, $Intercept_{MB}$ is more important than $d_{MB}$ is. From Figure 7, it is easy to see that all the four QSM-based feature vectors are functional in classifying tree species. Comparatively speaking, the relative importance of $L_{12}$ is lower than that of the other three QSM-based feature vectors. To reduce the feature vector dimension, five feature vectors with a low level of relative importance in these three kinds of feature vectors were ignored in the following SVM classification. These five feature vectors are $\kappa, \delta$ and *DBH* in the directly measured feature vectors, $d_{MB}$ in the fractal geometry-based feature vectors and $L_{12}$ in the QSM-based feature vectors.

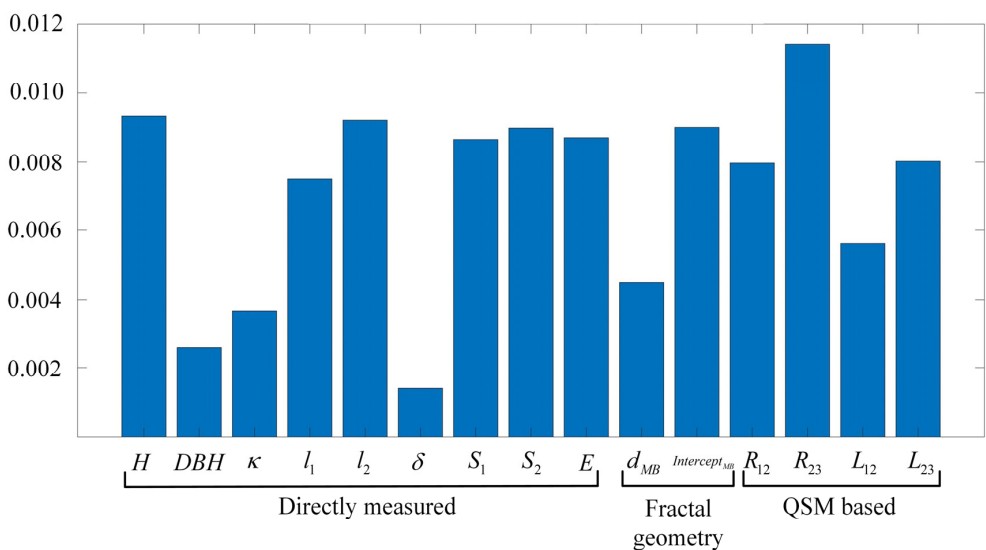

**Figure 7.** Relative importance of each vector calculated using the CART.

### 2.2.3. Selection of Feature Vectors Combination

After dimensionality reduction, ten feature vectors were retained for further processing. To select the final combination of feature vectors, 4-fold cross validation was conducted via applying different combination of feature vectors with different dimensions. The 4-fold cross validation method is shown in Figure 8. The datasets were classified into four sets. Three sets were selected for training, while the remaining one was used for testing. The mean accuracy for the four results is given as the classification accuracies.

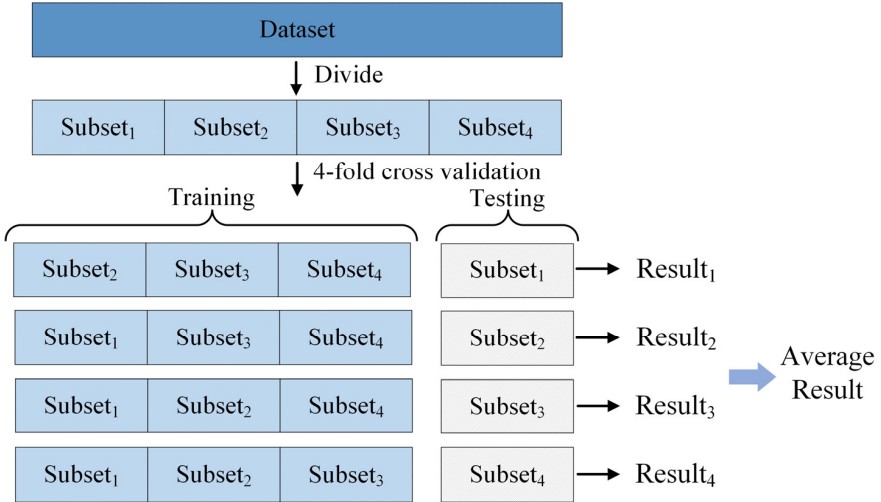

**Figure 8.** Fourfold cross validation.

The results of using different combination of feature vectors are tabulated in Table 2. It can be found that the dimensionality of the feature vectors changes from 1 to 10. Different dimensionality of the feature vectors contains a different number of combination of feature vectors. For instance, when the dimensionality is 1, there are 10 different combinations, such as $\{H\}$, $\{l_1\}$ and $\{L_{23}\}$, etc. Although the dimensionality is same, different combinations of feature vectors have different classification accuracies. From Table 2, it is easy to find that when the dimensionality is 3, the lowest overall accuracy of the combination of feature vectors is 0.421, while the highest overall accuracy is 0.845. In total, there are 1023 different combinations of feature vectors. In general, if one feature vector is noticeably effective at tree species classifying, the occurrence frequency of this feature vector should be high among the combinations with a high classifying accuracy. Thus, to further analysis the relative importance of each feature vector, the occurrence frequency of each feature vector among the different combinations of feature vectors whose classifying accuracy is higher than 0.85 was statistically counted.

**Table 2.** Overall accuracy of different combination of feature vectors.

| Dimensionality | Lowest Overall Accuracy | Highest Overall Accuracy | Combination of Feature Vectors | Number of Combinations |
|---|---|---|---|---|
| 1 | 0.354 | 0.583 | $\{H\},\{l_1\}\ldots\{L_{23}\}$ | 10 |
| 2 | 0.426 | 0.734 | $\{H,l_1\},\{H,l_2\}\ldots\{R_{23},L_{23}\}$ | 45 |
| 3 | 0.421 | 0.845 | $\{H,l_1,l_2\},\{H,l_1,S_1\}\ldots\{R_{12},R_{23},L_{23}\}$ | 120 |
| 4 | 0.440 | 0.896 | $\{H,l_1,l_2,S_1\},\{H,l_1,l_2,S_2\}\ldots\{Intercept_{MB},R_{12},R_{23},L_{23}\}$ | 210 |
| 5 | 0.521 | 0.938 | $\{H,l_1,l_2,S_1,S_2\},\{H,l_1,l_2,S_1,E\}\ldots\{E,Intercept_{MB},R_{12},R_{23},L_{23}\}$ | 252 |
| 6 | 0.610 | 0.928 | $\{H,l_1,l_2,S_1,S_2,E\},$ $\{H,l_1,l_2,S_1,S_2,Intercept_{MB}\}\ldots\{S_2,E,Intercept_{MB},R_{12},R_{23},L_{23}\}$ | 210 |
| 7 | 0.688 | 0.931 | $\{H,l_1,l_2,S_1,S_2,E,Intercept_{MB}\},$ $\{H,l_1,l_2,S_1,S_2,E,R_{12}\}\ldots\{S_1,S_2,E,Intercept_{MB},R_{12},R_{23},L_{23}\}$ | 120 |
| 8 | 0.783 | 0.947 | $\{H,l_1,l_2,S_1,S_2,E,Intercept_{MB},R_{12}\},$ $\{H,l_1,l_2,S_1,S_2,E,Intercept_{MB},R_{23}\}\ldots\{l_2,S_1,S_2,E,Intercept_{MB},R_{12},R_{23},L_{23}\}$ | 45 |
| 9 | 0.880 | 0.938 | $\{H,l_1,l_2,S_1,S_2,E,Intercept_{MB},R_{12},R_{23}\},$ $\{H,l_1,l_2,S_1,S_2,E,Intercept_{MB},R_{12},L_{23}\}\ldots\{l_1,l_2,S_1,S_2,E,Intercept_{MB},R_{12},R_{23},L_{23}\}$ | 10 |
| 10 | 0.923 | 0.923 | $\{H,l_1,l_2,S_1,S_2,E,Intercept_{MB},R_{12},R_{23},L_{23}\}$ | 1 |
| In total | / | / | / | 1023 |

## 3. Experimental Result and Analysis

### 3.1. Feature Vector Dimensional Reduction

As mentioned above, to further analyze the relative importance of each feature vector, the occurrence frequency of each feature vector was statistically counted. The statistical result is shown in Figure 9. It can be found that the occurrence frequency of tree height ($H$) feature vector is the highest. Comparatively speaking, the minor axis of the canopy ellipse ($l_2$) and canopy convex hull area ($S_1$) have the lowest frequency. These two feature vectors were ignored in the final SVM classification. Thus, eight feature vectors were actually selected for tree species classification. The reason for this can be explained as follows.

As can be seen in Table 2, when the feature vector dimension is less than eight, its lowest or highest overall accuracies are all smaller than those of the feature sets whose dimension is eight. In addition, when the feature vector dimension is larger than eight, its highest overall accuracy is still smaller than that of the feature sets whose dimension is eight. Moreover, a smaller feature vector dimension means that fewer calculations are

involved. Thus, the final eight feature vectors, including $H$, $R_{23}$, $L_{23}$, $Intercept_{MB}$, $R_{12}$, $E$, $S_2$ and $l_1$ were adopted for tree species classification in this paper.

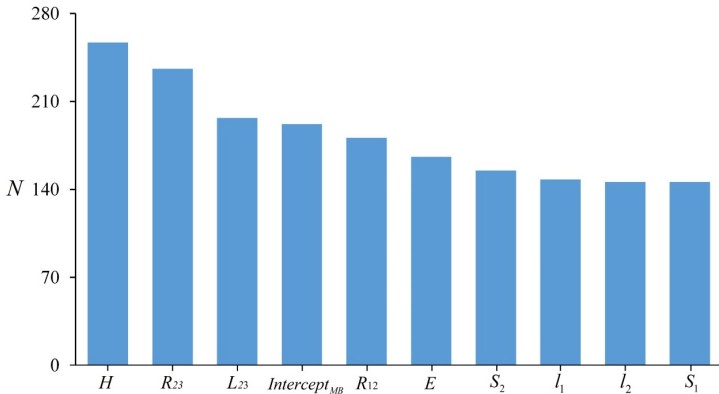

**Figure 9.** Occurrence frequency of each feature vector.

*3.2. Experimental Results*

Five accuracy indexes were adopted for evaluating the performance of the proposed method, including the overall accuracy (*OA*), precision (*Pre*), recall (*Rec*), F1 score ($F_1$) and kappa coefficient (*Ka*). These five accuracy indexes are defined as Equations (8)–(13), respectively:

$$OA = \frac{TP + TN}{TP + TN + FP + FN} \tag{8}$$

$$Pre = \frac{TP}{TP + FP} \tag{9}$$

$$Rec = \frac{TP}{TP + FN} \tag{10}$$

$$F_1 = 2 \times \frac{Pre \times Rec}{Pre + Rec} \tag{11}$$

$$Ka = \frac{OA - Pe}{1 - Pe} \tag{12}$$

$$Pe = \frac{(TP + FN) \times (TP + FP) + (FP + TN) \times (FN + TN)}{N^2} \tag{13}$$

where $TP$ is the number of trees correctly classified as the specific species, $TN$ is the number of trees correctly classified as other species, $FN$ is the number of trees misclassified as other species and $FP$ is number of trees misclassified as the specific species by other species. The confusion matrix of the classification results in this paper is shown in Figure 10.

| Result / Reference | FagSyl | PciAbi | PinSyl | PseMen | QuePet |
|---|---|---|---|---|---|
| FagSyl | 126 | 0 | 1 | 2 | 0 |
| PciAbi | 0 | 115 | 0 | 0 | 8 |
| PinSyl | 3 | 0 | 74 | 4 | 0 |
| PseMen | 4 | 0 | 5 | 115 | 0 |
| QuePet | 1 | 10 | 0 | 0 | 100 |

**Figure 10.** Confusion matrix of the classification results in this paper.

According to the classification results shown in Figure 10, the five accuracy indexes were calculated, as tabulated in Table 3. In terms of $OA$, all the tree species have a higher classification overall accuracy, whose $OA$ are all high than 95%. In terms of *Pre*, the precision values of all the tree species are all larger than 90%. This means that the proposed method can obtain satisfactory classification results for all the five tree species. In Figure 10, it can be seen that there are 134 trees whose tree species are fagus sylvatica. In the classification results, 126 trees of fagus sylvatica were correctly classified. In terms of other accuracy indexes, such as $F_1$ and $Ka$, the proposed method can all obtain satisfactory classification results. Thus, it can be concluded that the developed feature vectors have a remarkable effect on tree species classification.

**Table 3.** Accuracy metrics calculation results for the five different tree species.

|  | FagSyl | PicAbi | PinSyl | PseMen | QuePet |
|---|---|---|---|---|---|
| $OA$ (%) | 98.06 | 96.83 | 97.71 | 97.36 | 96.65 |
| $Pre$ (%) | 94.03 | 92.00 | 92.50 | 95.04 | 92.59 |
| $Rec$ (%) | 97.67 | 93.50 | 91.36 | 92.74 | 90.09 |
| $F_1$ (%) | 95.82 | 92.74 | 91.93 | 93.88 | 91.32 |
| $Ka$ (%) | 94.56 | 90.72 | 90.59 | 92.19 | 89.25 |

In addition to SVM, there are many other classical supervised learning methods, such as Adaptive Boosting (AdaBoost), K-Nearest Neighbors (KNN), Naive Bayes (NB) and random forest (RF). The authors of this paper further compared the accuracy metrics of tree species classification via SVM with the ones of other supervised learning methods. The comparison results are shown in Table 4. In Table 4, it can be seen that the proposed method can obtain the best tree species classification results, no matter which accuracy index is adopted. In terms of $OA$, the proposed method achieved 93.31% overall accuracy, while the overall accuracies of the other four methods are all smaller than 90%. In terms of $F_1$ and $Ka$, the proposed method still performed the best. Thus, it can be concluded that SVM adopted in this paper outperformed the other four classical supervised learning methods in terms of tree species classification.

**Table 4.** Accuracy metrics of different supervised learning methods.

|  | AdaBoost | KNN | NB | RF | The Proposed Method |
|---|---|---|---|---|---|
| $OA$ (%) | 88.03 | 89.61 | 81.34 | 89.79 | 93.31 |
| $Pre$ (%) | 87.69 | 89.63 | 81.11 | 89.66 | 93.23 |
| $Rec$ (%) | 87.34 | 89.21 | 81.37 | 88.95 | 93.07 |
| $F_1$ (%) | 87.49 | 89.37 | 80.95 | 89.22 | 93.14 |
| $Ka$ (%) | 85.36 | 87.26 | 77.60 | 87.44 | 91.72 |

## 4. Discussion

In this paper, the dimensionality of feature vectors was reduced from fifteen to eight. In achieving this, the computation burden can be relieved. The final feature vectors combination contains $H$, $R_{23}$, $L_{23}$, $Intercept_{MB}$, $R_{12}$, $E$, $S_2$ and $l_1$, according to the occurrence frequency of each feature vector. As mentioned above, this paper provides a strategy for feature dimension reduction. To further discuss the performance of the reduced feature vectors, the authors of this paper tested all the tree species classification results using all the feature vector combinations when the feature dimension was eight. The results are shown in Table 5. In Table 5, it can be seen that there are forty-five different feature vector combinations in total. In terms of $OA$, there are only four combinations whose $OA$ is a little larger than the feature vector combination adopted in this paper was. Four combinations are colored in blue, as shown in Table 5. It is easy to see that their $OA$s are less than one

percentage points higher than that of the proposed eight feature vectors. Meanwhile, the performances of the proposed eight feature vectors are better than the ones of all the other forty-feature vector combinations, no matter which accuracy indicator was adopted. Thus, it can be concluded that using the occurrence frequency of each feature vector will provide an effective dimension reduction method for tree species classification, while maintaining a promising tree species classification performance.

**Table 5.** Accuracy metrics comparison of different combination of feature vectors. (The background color of the combination of feature vectors adopted in this paper was red, while the background color of four feature vector combinations, whose performances are a little better than the proposed feature vector combinations was blue.)

| Combination | OA (%) | Pre (%) | Rec (%) | $F_1$ (%) | Ka (%) |
|---|---|---|---|---|---|
| $\{H,l_1,l_2,S_1,S_2,E,Intercept_{MB},R_{12}\}$ | 85.74 | 84.99 | 85.19 | 85.05 | 82.08 |
| $\{H,l_1,l_2,S_1,S_2,,Intercept_{MB},R_{23}\}$ | 88.73 | 87.83 | 88.01 | 87.9 | 85.85 |
| $\{H,l_1,l_2,S_1,S_2,E,Intercept_{MB},L_{23}\}$ | 82.04 | 80.79 | 81.18 | 80.82 | 77.44 |
| $\{H,l_1,l_2,S_1,S_2,E,R_{12},R_{23}\}$ | 85.74 | 84.71 | 84.63 | 84.66 | 82.06 |
| $\{H,l_1,l_2,S_1,S_2,E,R_{12},L_{23}\}$ | 84.86 | 84.22 | 84.64 | 84.39 | 80.98 |
| $\{H,l_1,l_2,S_1,S_2,E,R_{23},L_{23}\}$ | 89.61 | 89.45 | 89.29 | 89.34 | 86.93 |
| $\{H,l_1,l_2,S_1,S_2,Intercept_{MB},R_{12},R_{23}\}$ | 88.38 | 87.62 | 87.65 | 87.62 | 85.39 |
| $\{,l_1,l_2,S_1,S_2,Intercept_{MB},R_{12},L_{23}\}$ | 88.38 | 87.98 | 88.26 | 88.09 | 85.4 |
| $\{H,l_1,l_2,S_1,S_2,Intercept_{MB},R_{23},L_{23}\}$ | 91.02 | 90.85 | 90.67 | 90.75 | 88.71 |
| $\{H,l_1,l_2,S_1,S_2,R_{12},R_{23},L_{23}\}$ | 90.32 | 90.16 | 89.91 | 90.00 | 87.82 |
| $\{H,l_1,l_2,S_1,E,Intercept_{MB},R_{12},R_{23}\}$ | 87.85 | 86.86 | 86.85 | 86.84 | 84.73 |
| $\{H,l_1,l_2,S_1,E,Intercept_{MB},R_{12},L_{23}\}$ | 87.68 | 87.39 | 87.25 | 87.26 | 84.5 |
| $\{H,l_1,l_2,S_1,E,Intercept_{MB},R_{23},L_{23}\}$ | 92.78 | 92.93 | 92.33 | 92.58 | 90.91 |
| $\{H,l_1,l_2,S_1,E,R_{12},R_{23},L_{23}\}$ | 93.13 | 92.85 | 92.86 | 92.85 | 91.37 |
| $\{H,l_1,,S_1,Intercept_{MB},R_{12},R_{23},L_{23}\}$ | 91.9 | 91.61 | 91.68 | 91.64 | 89.82 |
| $\{H,l_1,l_2,S_2,E,Intercept_{MB},R_{12},R_{23}\}$ | 89.44 | 88.65 | 88.61 | 88.61 | 86.72 |
| $\{H,l_1,l_2,S_2,E,Intercept_{MB},R_{12},L_{23}\}$ | 87.85 | 87.65 | 87.37 | 87.46 | 84.72 |
| $\{H,l_1,l_2,S_2,E,Intercept_{MB},R_{23},L_{23}\}$ | 93.13 | 93.44 | 92.82 | 93.08 | 91.36 |
| $\{H,l_1,l_2,S_2,E,R_{12},R_{23},L_{23}\}$ | 91.02 | 90.54 | 90.8 | 90.65 | 88.72 |
| $\{H,l_1,l_2,S_2,Intercept_{MB},R_{12},R_{23},L_{23}\}$ | 92.43 | 92.24 | 92.28 | 92.23 | 90.48 |
| $\{H,l_1,l_2,E,Intercept_{MB},R_{12},R_{23},L_{23}\}$ | 92.25 | 92.08 | 92.1 | 92.07 | 90.26 |
| $\{H,l_1,S_1,S_2,E,Intercept_{MB},R_{12},R_{23}\}$ | 88.56 | 87.98 | 87.86 | 87.89 | 85.61 |
| $\{H,l_1,S_1,S_2,E,Intercept_{MB},R_{12},L_{23}\}$ | 89.79 | 89.37 | 89.58 | 89.46 | 87.17 |
| $\{H,l_1,S_1,S_2,E,Intercept_{MB},R_{23},L_{23}\}$ | 94.01 | 94.23 | 93.89 | 93.96 | 92.47 |
| $\{H,l_1,S_1,S_2,E,R_{12},R_{23},L_{23}\}$ | 91.37 | 91.03 | 91.14 | 91.05 | 89.16 |
| $\{H,l_1,S_1,S_2,Intercept_{MB},R_{12},R_{23},L_{23}\}$ | 90.67 | 90.48 | 90.32 | 90.38 | 88.26 |
| $\{H,l_1,S_1,E,Intercept_{MB},R_{12},R_{23},L_{23}\}$ | 94.19 | 94.12 | 93.94 | 94.02 | 92.69 |
| $\{H,l_1,S_2,E,Intercept_{MB},R_{12},R_{23},L_{23}\}$ | 93.31 | 93.23 | 93.07 | 93.14 | 91.72 |
| $\{H,l_2,S_1,S_2,E,Intercept_{MB},R_{12},R_{23}\}$ | 89.61 | 88.95 | 89.13 | 89.03 | 86.95 |
| $\{H,l_2,S_1,S_2,E,Intercept_{MB},R_{12},L_{23}\}$ | 90.14 | 90.11 | 89.72 | 89.86 | 87.6 |
| $\{H,l_2,S_1,S_2,E,Intercept_{MB},R_{23},L_{23}\}$ | 91.9 | 91.93 | 91.61 | 91.74 | 89.81 |
| $\{H,l_2,S_1,S_2,E,R_{12},R_{23},L_{23}\}$ | 90.67 | 90.39 | 90.49 | 90.39 | 88.27 |
| $\{H,l_2,S_1,S_2,Intercept_{MB},R_{12},R_{23},L_{23}\}$ | 91.9 | 91.7 | 91.78 | 91.65 | 89.83 |
| $\{H,l_2,S_1,E,Intercept_{MB},R_{12},R_{23},L_{23}\}$ | 93.84 | 93.81 | 93.75 | 93.78 | 92.25 |
| $\{H,l_2,S_2,E,Intercept_{MB},R_{12},R_{23},L_{23}\}$ | 93.66 | 93.39 | 93.33 | 93.35 | 92.03 |
| $\{H,S_1,S_2,E,Intercept_{MB},R_{12},R_{23},L_{23}\}$ | 91.37 | 91.07 | 91.27 | 91.14 | 89.16 |
| $\{l_1,l_2,S_1,S_2,E,Intercept_{MB},R_{12},R_{23}\}$ | 81.51 | 81.09 | 81.05 | 81.04 | 76.76 |
| $\{l_1,l_2,S_1,S_2,E,Intercept_{MB},R_{12},L_{23}\}$ | 82.92 | 82.77 | 82.53 | 82.62 | 78.52 |
| $\{l_1,l_2,S_1,S_2,E,Intercept_{MB},R_{23},L_{23}\}$ | 85.21 | 85.88 | 85.66 | 85.71 | 81.4 |
| $\{l_1,l_2,S_1,S_2,E,R_{12},R_{23},L_{23}\}$ | 80.46 | 80.99 | 80.69 | 80.8 | 75.42 |
| $\{l_1,l_2,S_1,S_2,Intercept_{MB},R_{12},R_{23},L_{23}\}$ | 85.21 | 86.1 | 85.91 | 85.99 | 81.4 |
| $\{l_1,l_2,S_1,E,Intercept_{MB},R_{12},R_{23},L_{23}\}$ | 88.03 | 88.37 | 88.16 | 88.23 | 84.94 |
| $\{l_1,l_2,S_2,E,Intercept_{MB},R_{12},R_{23},L_{23}\}$ | 86.27 | 86.92 | 86.7 | 86.75 | 82.73 |
| $\{l_1,S_1,S_2,E,Intercept_{MB},R_{12},R_{23},L_{23}\}$ | 86.27 | 86.72 | 86.74 | 86.67 | 82.74 |
| $\{l_2,S_1,S_2,E,Intercept_{MB},R_{12},R_{23},L_{23}\}$ | 86.27 | 86.73 | 86.64 | 86.62 | 82.73 |

Akerblom et al. [6] developed feature vectors using the QSM for recognizing tree species. In their method, three different types of feature vectors, namely the stem branch, crown and tree, were presented. There were fifteen feature vectors in total, including the stem branch angle, stem branch cluster size, three stem branches' radius, stem branch length, stem branch distance, crown start height, crown height, crown evenness, crown diameter/height, DBH/height ratio, DBH/tree volume, DBH/minimum tree radius, volume below 55% of the total height, cylinder length/tree volume and shedding ratio. Three different classification methods, namely KNN, multinomial regression (MNR) and SVM (including three different kernel functions, such as linear, polynomial and radial basis functions, which are $SVM_{lin}$, $SVM_{pol}$ and $SVM_{rbf}$, respectively) were applied for tree species classification. The authors of this paper compared their performance with that of the proposed method. The comparison is shown in Table 6. In Table 6, it can be seen that compared with the fifteen feature vectors developed by Akerblom et al. [6], although only eight feature vectors were adopted in this paper, the proposed method obtained much better tree species classification results. Thus, it can be concluded that the eight feature vectors developed in this paper are more effective at tree species classification.

**Table 6.** Accuracy metrics comparison with the method proposed by Åkerblom et al. [6].

| | Åkerblom et al. [6] | | | | | The Proposed Method |
|---|---|---|---|---|---|---|
| | **KNN** | **MNR** | $SVM_{lin}$ | $SVM_{pol}$ | $SVM_{rbf}$ | |
| *OA* (%) | 79.75 | 81.87 | 81.69 | 75.88 | 82.04 | 93.31 |
| *Pre* (%) | 80.00 | 80.94 | 80.87 | 78.79 | 81.82 | 93.23 |
| *Rec* (%) | 78.18 | 80.93 | 80.96 | 73.54 | 80.75 | 93.07 |
| $F_1$ (%) | 78.44 | 80.90 | 80.91 | 74.01 | 81.09 | 93.14 |
| *Ka* (%) | 74.46 | 77.21 | 76.99 | 69.47 | 77.37 | 91.72 |

## 5. Conclusions

Tree species have a direct impact on forests' productivity and diversity. To accurately and efficiently identify tree species, this paper proposed a tree species classification method based on the combination of fractal geometric feature vectors and QSM feature vectors. In this paper, three different types of feature vectors were first extracted. Specially, fractal geometry-based feature vectors, including fractal dimension and intercept, were developed. Meanwhile, four different QSM feature vectors were also introduced to obtain better tree species classification results. Successively, the feature vector dimensionality was reduced by analyzing the relative importance for tree species identification using the CART method. To further conduct feature dimension reduction, the occurrence frequency of each feature vector was statistically calculated using different combinations of feature vectors. Eventually, SVM was applied to obtain tree species classification results using the reduced eight feature vectors. Five hundred and sixty-eight individual tree point clouds with five tree species were used for testing. The experimental results show that the developed feature vectors are effective at identifying different tree species. The overall accuracies for the five tree species are all greater than 95%. In the comparison with the other four classical supervised learning methods, the proposed method still performs the best. Compared with the relevant method, the eight feature vectors developed in this paper also performed much better. This indicates that the fractal geometry-based feature vectors and QSM-based feature vectors developed in this paper can effectively improve the performance of tree species classification.

**Author Contributions:** Conceptualization, Z.H.; experimental analysis, Z.C. and P.C.; writing—original draft preparation, Z.H. and Z.C.; writing—review and editing, Y.X. and P.X. All authors have read and agreed to the published version of the manuscript.

**Funding:** This work was supported by the National Natural Science Foundation of China (NSF) (42161060, 41801325, 41861052, 42174055 and 41962018), Research on Key technologies of spatio-

temporal information and collaborative design in engineering (YJZD2020-01), the China Post-Doctoral Science Foundation (2019M661858), the Natural Science Foundation of Jiangxi Province (20192BAB217010) and East China University of Technology's Ph.D. Project (DHBK2017155) for their financial support.

**Data Availability Statement:** Not applicable.

**Conflicts of Interest:** The authors declare no conflict of interest.

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
