# Peer review of "Tree Species Classification Using Optimized Features Derived from Light Detection and Ranging Point Clouds Based on Fractal Geometry and Quantitative Structure Model"

_forests, doi:10.3390/f14061265_

Round 1

Reviewer 1 Report

The manuscript is about tree species classification based on LiDAR point cloud data. As such the topic is interesting and important for the readers of Forests. The idea of using LiDAR data and QSMs derived from the data is not new and the present work is more about development of the idea by expanding the types of features used. In this paper the authors used features directly computed from the point cloud such as tree height and projected crown area, QSM derived features such as ratios of 2nd-order branch radius to 1st-order branch radius, and the fractal dimension. The fractal dimension is the novelty as such and the specific QSM derived features are also new ones. Another somewhat novel aspect is the feature selection where the authors use the CART method to rank the features to select 8 optimal features from the 15 features defined. The work is a useful but perhaps quite modest addition to the literature. The paper can be accepted for publication after some revisions.

Major comments:

Authors write in many places how the number of features needs to be small to reduce the computational burden (e.g. 369-370: “To alleviate the computational burden, the feature vector dimensionality was reduced”). But there are no mention about some kind of online or realtime data processing and classification need. Also, if the computational burden is so big an issue, which I don’t see it to be, there should be lots of statistics presented about the computational time, memory requirement, etc for the classification tasks with different number of features, types of features, on classification methods. And then discussion about these reduced computational requirements and their possible implications.

There are quite a lot of repetitions or unnecessary explanations that can be removed/shortened to make the paper shorter. For example, the 15 features and the 3 different types (point cloud, fractal, QSM) are mentioned and defined and tabulated many times. Similarly, the explantation of the QSM features is very tedious and long where the image 4 is easier to understand than the long written explanation.

One of the major aspects of the paper is the feature selection based on CART method. The authors should explain with much more detail how that works and how it computes those relative importance values shown in fig 5.  

Section 3.1 Experimental datasets should be in the Methods section (section 2.). These data are not only used for experiments in section 3 but also in section 2.2 Dimensionality reduction of feature vectors. Also, the dimensional reduction part, not the method but its results, should be part of the results section.

How was the number of 8 features actually selected? What was the reason to have 8 and not 7 or 9 features?

Some minor comments:

170-171: “Fractal geometry considers that many objects own the hierarchy of self-similarity, in an ideal world, even own infinite levels.” What does “even own infinite levels” mean?

Table 1.  Is this table really needed because all the features were already defined and classified into the 3 different types in the text and also in the figure 2? Thus maybe this table could be removed.

263-264: “It can be found that the occurrence frequency of tree height (H) feature vector is the highest. It means H can make more contribution to tree species classification.” What do you mean exactly that the height “can make more contribution to tree species classification”? The fact that height was the most frequent feature in your test does in itself mean that it makes or can make more contribution to the classification.

Table 5. The prposed method —> the proposed method

347-348 “It means this combination of feature vectors can achieve better tree species classification results when the dimensionality of feature vectors is eight. “ This is not a true statement in general while it might be true in some cases. Just because these eight features were the most frequent in the earlier test, does not mean their combination is the best for the classification. For example, it might be that two of the features are quite correlated in which case using both of them does not improve the classification much at all. So please rephrase this.

356-357: “In terms of OA , the lowest value is 66.20%, while the optimal combination obtained 27.11% higher overall accuracy. “  The highest OA is not 27.11 % higher than the lowest. It is 27.11 percentage points higher or 41% higher.

The language is ok but could be improved, for example some sentences could be shortened and made less complicated.

Reviewer 2 Report

The abstract provides an overview of a research paper on tree species classification using LiDAR technology. However, there are a few areas where more information could be included:

1. Research gap: The abstract mentions that developing effective feature vectors for tree species classification while maintaining low computation burden is a challenge. It would be helpful to briefly explain why the existing methods or feature vectors are insufficient and how the proposed method addresses this gap.

2. Methodology: The abstract mentions the use of fractal geometry and quantitative structural model (QSM) construction for tree species classification. It would be beneficial to provide a brief explanation of these methods and how they are applied in the study. Additionally, more details about the specific techniques used to develop the feature vectors based on fractal geometry and QSM could be included.

3. Evaluation: The abstract states that 568 individual trees with five tree species were selected to evaluate the performance of the developed feature vectors. It would be useful to mention how the evaluation was conducted, such as the training and testing procedures, performance metrics used, and any statistical analyses performed.

4. Results: While the abstract mentions that the proposed method achieved an overall accuracy above 95% for all five tree species, it would be helpful to provide more specific details about the performance. For example, including the accuracy values for each individual tree species or discussing any variations in performance across different species.

5. Comparison: The abstract mentions that the proposed method outperforms four classical supervised learning methods, but it does not provide any details on the specific methods or the basis for comparison. Adding a sentence or two to briefly describe the compared methods and the results of the comparison would give readers a better understanding of the proposed method's superiority.

6. Conclusion: The abstract does not include a concluding statement summarizing the main findings or the significance of the research. Adding a brief concluding sentence would help readers understand the key takeaways of the study.

Including these additional elements would enhance

Fig 6 can be improved

The discussion section can be extended by accuracy metrics that can be compared with relevant literature.

Overall, revision require.

Round 2

Reviewer 1 Report

The authors have quite well replied to the comments and modified the manuscript accordingly. The paper can be accepted for publication after some minor modifications.

Minor comments:

Abstract: The new abstract mentions the machine learning methods that were compared (e.g. KNN, NB) but not the SVM method that was the proposed method. You should describe the proposed method better in the abstract. But the abstract, while very informative and well written, is now very long and I would encourage the authors to try shortening it.

Equation 7: You should define in the text what is “D” in “”Gini(D) = …”. You also talk about “nodes” in the explanation of the Gini index and how there is risk of splitting. Could you define what these nodes are in your case and what is this risk of splitting.

Discussion: 373-389: “As mentioned above, the optimal combination of feature vectors contains H, R23 , L23 , InterceptMB , R12 , E , S2 and l1 according to the occurrence frequency of each feature vector. It means that when feature vector dimension is eight, the feature set with highest occurrence frequency will outperform other feature sets whose feature vector dimension is eight for tree species classification. From Table 2, it can be found that there are 45 different combinations of feature vectors when feature vectors dimensionality is eight. To verify the effectiveness of the optimal combination of feature vector, this paper randomly selected eight feature vectors from Table 2 and compared their classification performance with that of the optimal combination of feature vectors. The comparison results are shown in Table 5. It is easy to find that the optimal combination of feature vectors extracted by this paper obtained the best classification results no matter which accuracy index is adopted. In terms of OA , the lowest value is 66.20%, while the optimal combination obtained 27.11 percentage points higher overall accuracy. From Table 6, it can also be found that the classification accuracy of different feature vectors varies greatly from each other. Compared with other combination of feature vectors, the optimal combination adopted in this paper achieved the best performance no matter which accuracy index is selected.” I want to repeat that while these 8 features had the highest occurrence frequency in the earlier test, it does not logically follow that they are the optimal combination of 8 features, i.e. the best 8 feature combination. To show this you should test all the 45 different combinations, which is not that many, and not just 8 randomly selected combinations. Consider testing all the 45 combinations and showing a sample of them as in Table 5. “From Table 6…” should be “From Table 5 …”?
